# Track Reconstruction in a High-Density Environment with ALICE

**Mesut Arslandok** [1,2,3,*] , **Ernst Hellbär** [4] , **Marian Ivanov** [4,*] , **Robert Helmut Münzer** [5] and **Jens Wiechula** [5]

1   European Organization for Nuclear Research (CERN), 1211 Geneva, Switzerland
2   Wright Lab, Physics Department, Yale University, New Haven, CT 06520, USA
3   Physikalisches Institut, Ruprecht-Karls-Universität Heidelberg, 69120 Heidelberg, Germany
4   GSI Helmholtzzentrum für Schwerionenforschung, 64291 Darmstadt, Germany; ernst.hellbar@cern.ch
5   Institut für Kernphysik, Johann Wolfgang Goethe-Universität Frankfurt, 60438 Frankfurt, Germany; robert.muenzer@cern.ch (R.H.M.); jens.wiechula@cern.ch (J.W.)
*   Correspondence: mesut.arslandok@cern.ch (M.A.); marian.ivanov@cern.ch (M.I.)

**Abstract:** ALICE is the dedicated heavy-ion experiment at the CERN Large Hadron Collider (LHC). Its main tracking and particle-identification detector is a large volume Time Projection Chamber (TPC). The TPC has been designed to perform well in the high-track density environment created in high-energy heavy-ion collisions. In this proceeding, we describe the track reconstruction procedure in ALICE. In particular, we focus on the two main challenges that were faced during the Run 2 data-taking period (2015–2018) of the LHC, which were the baseline fluctuations and the local space charge distortions in the TPC. We present the corresponding solutions in detail and describe the software tools that allowed us to circumvent these challenges.

**Keywords:** heavy-ion collisions; tracking; Time Projection Chamber (TPC); space charge distortions; event pileup; interactive visualization

## 1. Introduction

ALICE was designed to cope with about 20,000 charged primary and secondary tracks emerging in the TPC acceptance ($|\eta| < 0.9$) from central Pb–Pb collisions at $\sqrt{s_{NN}}$ = 5.5 TeV. Such high-track densities are also achieved in pp collisions collected at high interaction rates, which cause pileups of particles from several collisions in the TPC drift time. The TPC is capable of tracking particles from very low ($\approx$100 MeV/$c$) up to fairly high ($\approx$100 GeV/$c$) transverse momentum ($p_T$). The apparatus consists of a central barrel enclosed in a solenoid magnet with a nominal field of 0.5 T along the beam direction, a forward muon spectrometer and several smaller detectors in the forward region [1,2].

In this proceeding, we will focus on the so-called combined tracking procedure, which uses the central barrel detectors; the Inner Tracking System (ITS), the Time Projection Chamber (TPC), which is the main tracking detector, the Transition Radiation Detector (TRD) and the Time-Of-Flight (TOF). The track reconstruction is based on the Kalman Filter approach [3,4], which consists of three steps as depicted in Figure 1. The tracking starts with the track seeding in the outermost pad rows of the TPC. The seed is then propagated inwards towards the primary vertex through the TPC volume and the ITS layers. Then, a second propagation step is performed in the outward direction from the innermost ITS layer to the outer detectors (TRD and TOF).

Finally, the primary tracks are refitted back to the primary vertex or as close to the vertex as possible in the case of secondary tracks. The improvement in $p_T$ resolution after applying a vertex constraint and including the TRD in the track fitting are shown in the left and right panels of Figure 2, respectively. The vertex constraint significantly improves the resolution of TPC standalone tracks, while it has no effect on ITS–TPC tracks (green and blue square overlap). Including TRD in the tracking improves the resolution by

about 40% at high $p_T$ for pp collisions recorded at both low (12 kHz) and high interaction (230 kHz) rates.

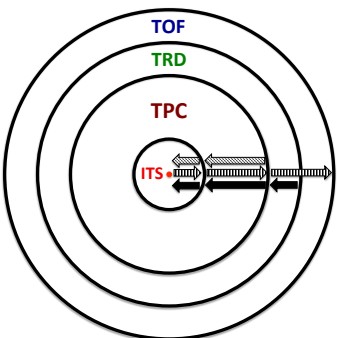

**Figure 1.** Schematic view of the three passes of the combined track finding [5,6]. Central barrel detectors are shown as cross-sections perpendicular to the beam direction from innermost (ITS) to outermost (TOF) detectors.

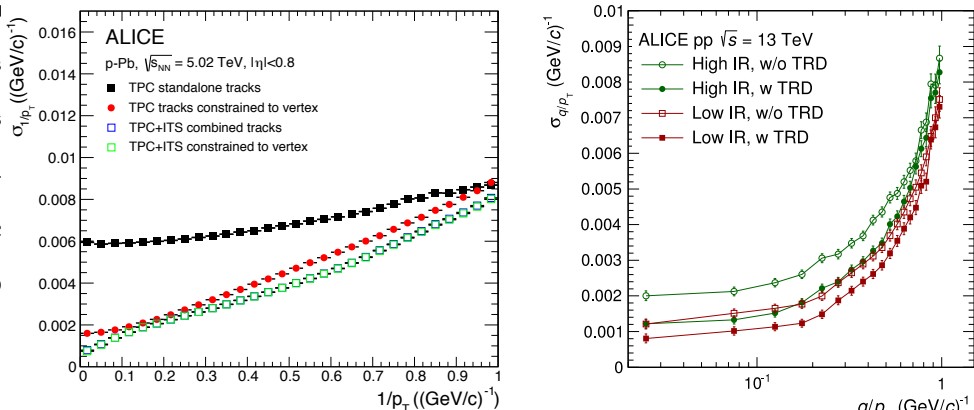

**Figure 2. Left:** the $p_T$ resolution in p–Pb collisions for standalone TPC and ITS–TPC matched tracks with and without constraint to the vertex [2]. **Right:** improvement of the $q/p_T$ (inverse transverse momentum scaled with the particle charge) resolution in data in pp collisions when TRD information is included in the tracking for various running scenarios. The labels low and high IR indicate interaction rates (IR) of 12 and 230 kHz, respectively [7].

## 2. Software Tools: Skimmed Data and RootInteractive

Tracking in a high-track density environment requires the efficient handling of large data samples and efficient software tools. The reconstructed Pb–Pb collision data collected by the ALICE detector until 2019 is on the level of several Petabytes. To process and calibrate such a large amount of data, we used the so-called "Skimmed data" approach, which provides a small-size data sample with sufficient statistics in all phase space containing all the required information. This is essential for a reliable and fast-turnaround optimization of the calibration and reconstruction algorithms. The data skimming procedure, which is depicted in orange and blue in Figure 3, allows for a substantial reduction of the disk space and processing time. More details will be given in the next paragraphs.

Input data, having a size on the order of several Petabytes, are skimmed down to several hundreds of Gigabytes. During this skimming, on the one hand, some of the tracks are skipped using a physics motivated down-scaling procedure (representative sampling of charged particles and $V^0$s (a $V^0$ is a neutral particle that decays into two charged tracks, such as $K_S^0$ ($K_S^0 \longrightarrow \pi^+\pi^-$), $\bar{\Lambda}$ ($\bar{\Lambda} \longrightarrow \bar{p}\pi^+$) and $\Lambda$ ($\Lambda \longrightarrow p\pi^-$)) depending on their $p_T$ and keeping the full track information, retaining nuclei and cosmic tracks etc.) and, thus, reducing the disk space.

On the other hand, additional information from different sources, such as the running conditions and derived variables obtained using the track and event properties are included

in the final tabulated summary data—the so-called "Skimmed Trees". These skimmed trees contain unbinned data. They are later converted to smaller n-dimensional histograms, which are subsequently saved into a tabulated tree (map) with a size on the order of a few Megabytes. These final trees preserve the full potential of the input collision data of several Petabytes.

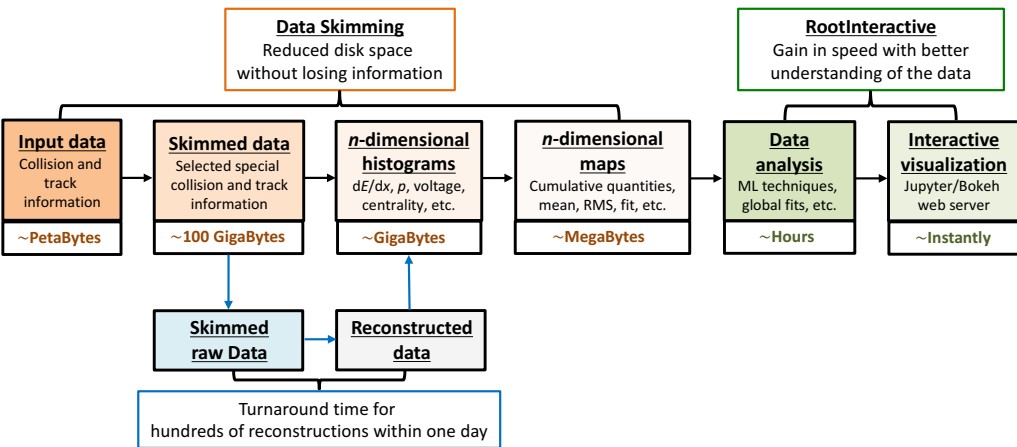

**Figure 3.** Workflow of the skimmed data and RootInteractive tool.

The final product of the skimming procedure described above is tabulated data in the TTree format, which can easily be converted into other formats, such as "panda", "numpy", "csv" etc. Eventually, it can be used as an input within the so-called "RootInteractive" framework [8]. One of the main advantages of RootInteractive is that it allows for interactive visualization of multidimensional data in ROOT or native Python formats. More importantly, by integrating Machine-Learning (ML) algorithms with interactive visualization tools, it makes data analysis more efficient and effective.

The graphical output of the tool is shown in Figure 4 for illustrative purposes. In this example, the dependencies of the common-mode effect (see Section 3) for the Gas Electron Multiplier (GEM)-based TPC [9] are studied using the Random Forest (RF) algorithm [10]. The tool allows for interactive visualization of eight parameters as sliders below the plots.

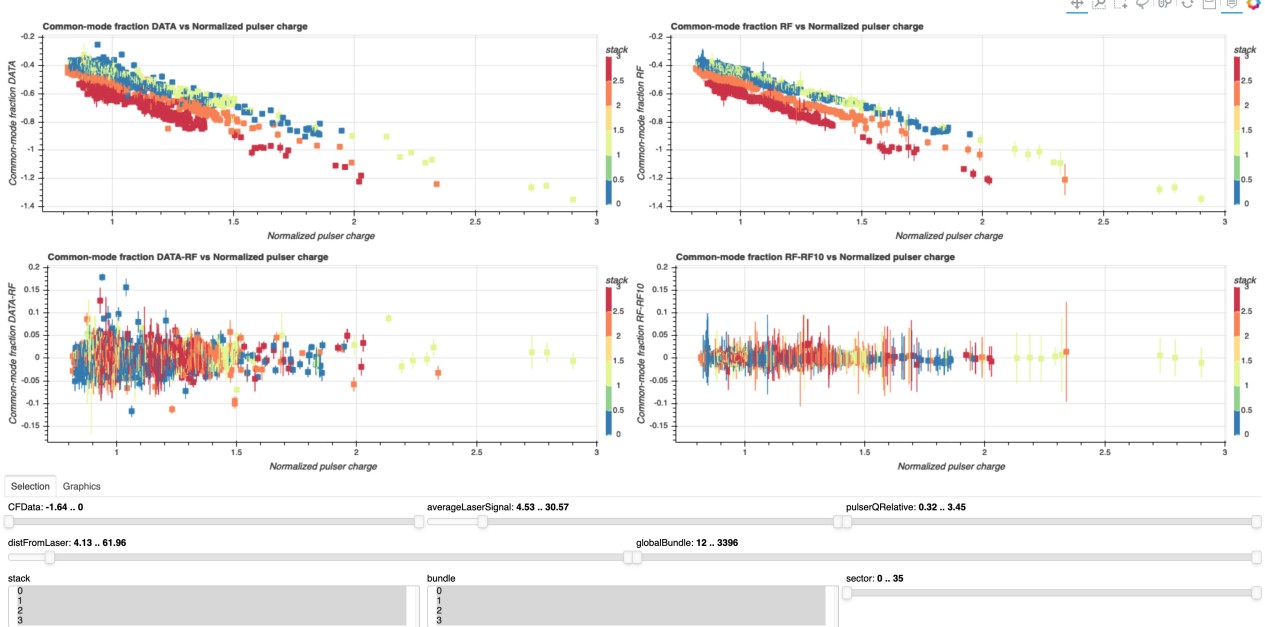

**Figure 4.** Illustration of the RootInteractive tool. Data (**top left**), comparison of the data to RF predictions (**top right**) and difference between the data and RF predictions (**bottom row**).

Raw data reconstruction and full Monte Carlo (MC) simulation use a large amount of CPU time. For instance, the reconstruction of the Pb–Pb data sample collected in 2018 took about five thousand years of CPU running time. Usually, the optimization of the reconstruction code is required to tune several parameters, thus, requiring running the reconstruction several times until the optimal performance is achieved. During the aforementioned data skimming, one can tag a sub-sample of events (high multiplicity, presence of nuclei, cosmic tracks etc.) that can be used for calibration purposes. Since the data size is substantially reduced by this tagging, we were able to run more than 200 different versions of the reconstruction code in parallel with a turnaround time of a single day.

### 3. Baseline Fluctuations in the TPC

The TPC signal has two characteristic features: the "ion-tail" and the "common-mode" effect. The signal induced on the readout pads of the TPC is characterized by a fast rise due to the ionization produced by the drifting electrons in the high electric field in the vicinity of the anode wire and a long ion-tail (more than 25% of the TPC drift time) due to the motion of back-drifting positive ions [11]. The magnitude of the negative undershoot caused by this ion-tail is usually smaller than 1% compared to the pulse height; however, its integral is about 50% of the integral of the signal itself. Therefore, in a high multiplicity environment, this ion-tail effect causes a significant degradation of the following signals on the same readout pad due to signal pileup.

Like the ion-tail, the common-mode effect also causes a multiplicity-dependent deterioration of the performance. This occurs due to the capacitive coupling of the anode wires to the readout pads. Due to this capacitive coupling, discharging and charging of the wires induces a bipolar signal on all pads facing the same anode wire segment in which the original signal is detected. The fast rise time of the discharging process causes a simultaneous undershoot. Figure 5 shows the ion-tail effect for different anode voltage settings (left panel) and two laser (to study the baseline effects and the signal shape, the Laser calibration system of the TPC was used [12]) track clusters on a given pad row, which induces common-mode signals on the other pads in the same time interval (right panel).

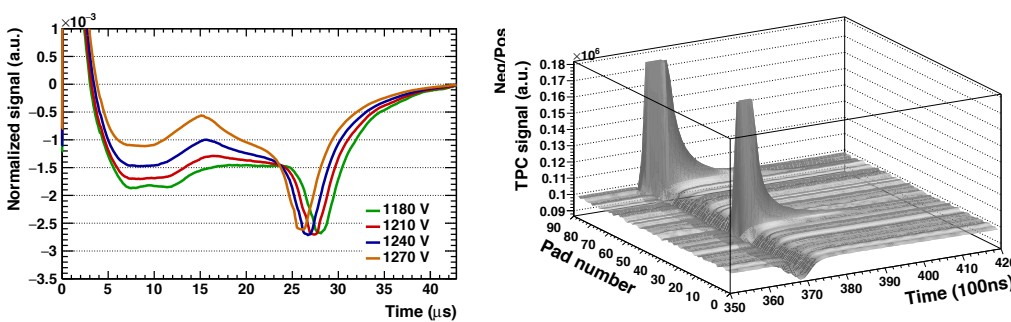

**Figure 5.** (Color online) **Left:** normalized pad signals with zoom into the y-axis showing the ion tail for different anode voltage settings. **Right:** laser track clusters (integrated over 2000 laser events) in a given pad-row illustrating the common-mode effect [13].

At high occupancy, both the ion-tail and the common-mode lead to a shift in the baseline. A toy simulation of a single pad readout in the presence of the ion-tail effect for a high-track multiplicity environment is illustrated in Figure 6. The performance of the baseline correction algorithm is shown in the right panel. In the Technical Design Report (TDR) of the TPC, it was assumed that the current front-end electronics will have a set of online signal processing algorithms (Moving Average Filter (MAF)) to correct for this baseline shift on the hardware level [14]. However, this functionality was not enabled due to instabilities in the firmware. Consequently, part of the pad signal remained under the zero-suppression threshold and was lost. This missing charge led to the loss of charge

within a cluster and, in some cases, also to a loss of clusters (e.g., when all pad signals of a cluster were below the zero suppression threshold).

Eventually, these baseline fluctuations resulted in a significant deterioration of the specific energy loss ($dE/dx$) measurement and consequently of the PID performance of the TPC, in particular, in a high-track density environment. Both of these effects were corrected offline during the data reconstruction and taken into account in the simulations at the digitization level.

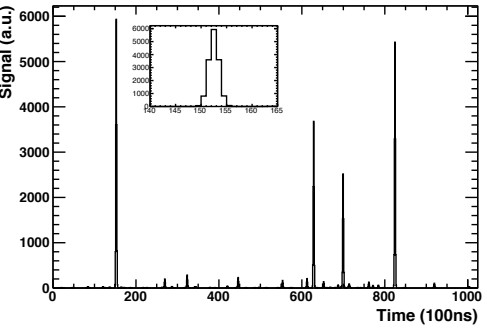 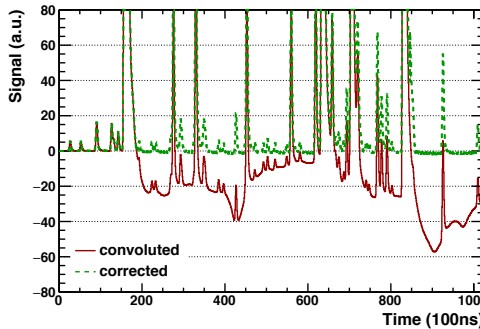

**Figure 6.** Toy simulation of a single pad readout at the presence of the ion-tail effect for a high-track multiplicity environment corresponding to a central Pb–Pb event at $\sqrt{s_{NN}}$ = 5.5 TeV. **Left:** input signal, where the zoom of the first peak along the time axis is shown in the inset. **Right:** convoluted (red solid line) and corrected (dashed green line) signals [13].

After 2015, the ALICE TPC was operated in Pb–Pb collisions at interaction rates of up to 8 kHz and pp collisions up to 200 MHz. This high interaction rate caused a pileup of interaction vertices within the TPC readout time and, consequently, a shift in the baseline of the readout electronics. The pileup of different collisions happens in two distinct natures; "same-bunch-crossing pileups", where two (or more) collisions occur in the same bunch crossing, and "out-of bunch pileups", where one (or more) collisions occur in bunch crossings different from the one that triggered the data acquisition.

In the first case, the collisions occur near in time with positions that are separated by up to few cm along the beam direction. These events can be identified based on multiple reconstructed vertices from tracks reconstructed in the TPC and ITS. In the case of out-of-bunch pileup, the collisions occur at different times, and therefore the tracks reconstructed in the TPC are spatially shifted along the drift direction (due to their different production times) and, in the vast majority of the cases not prolonged to the ITS, both due to the short readout time of ITS detectors and to their spatial shift in the TPC.

More than 25% of the Pb–Pb events collected in 2018 were influenced by out-of-bunch pileup collisions (while same-bunch pileup is negligible in Pb–Pb collisions at interaction rates of about 8 kHz). The resulting bias in the measured $dE/dx$ affects several physics analyses, for which the events with pileup collisions were discarded at the event selection level. Therefore, it is important to correct for the effect of pileup on $dE/dx$ to achieve optimal PID performance and avoid having to discard these events.

The left panel of Figure 7 shows the bias induced by pileup in the mean $dE/dx$ values for the pions belonging to the triggered collision. Depending on whether the pileup interaction occurred before or after the triggered interaction, the $dE/dx$ values systematically shift in a different direction. To correct for this effect, an event-by-event basis correction algorithm was developed.

Using the events with pileup, the deviation in the $dE/dx$ of tracks from the triggered collision was parameterized in four dimensions; pseudorapidity, TPC $dE/dx$, the multiplicity of the pileup event and the relative distance of the pileup collision vertex from the main interaction vertex along the beam axis. This multi-dimensional map was used to correct for the bias in the $dE/dx$ values, which was largely eliminated as shown in the right panel of Figure 7.

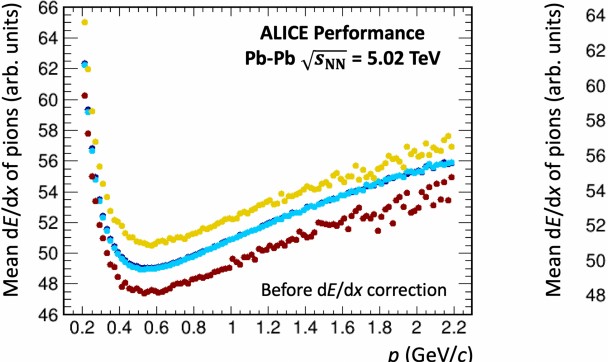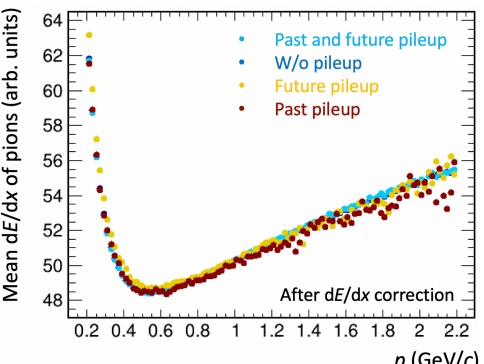

**Figure 7.** (Color online) Mean d$E$/d$x$ values of pions for the events with and without out-of-bunch pileup before (**left**) and after (**right**) d$E$/d$x$ correction.

As explained above, the common-mode effect and the ion-tail result in a significant loss of information. The reduction in the d$E$/d$x$ of the detected clusters is recovered by the offline correction mentioned above; however, the clusters that were lost because they were below the zero suppression threshold are not recoverable. Nevertheless, one can still account for this loss by increasing the cluster error by adding contributions stemming from baseline fluctuations. In this way, the efficiency of cluster-to-track assignment is partially restored. Since this problem mainly affects the particle detection efficiency, the full MC simulations must be treated accordingly.

Given its particular shape, the ion-tail produces a baseline bias that depends on pseudo-rapidity, d$E$/d$x$, momentum and track density. Therefore, a performance parametrization was carried out in multiple dimensions using the RootInteractive tool. Figure 8 shows the comparison of the Pb–Pb data and the corresponding MC productions in terms of the cluster finding efficiency before and after the baseline correction mentioned above. The observed discrepancy between the data and MC was significantly reduced by the baseline calibration. Note that calibration of the detector response in multiple dimensions is an optimization procedure of the overall performance. Therefore, slight residual imperfections for a given observable are inevitable.

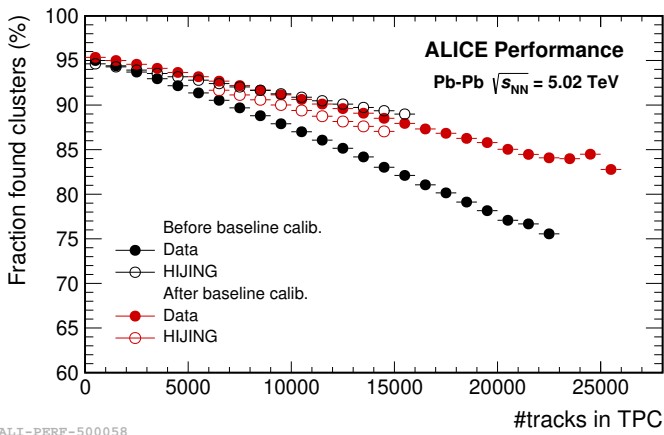

**Figure 8.** (Color online) Cluster finding efficiency as a function of track multiplicity before (black markers) and after (red markers) the baseline calibrations applied in the data and full simulation of the HIJING [15] event generator employing a GEANT4 implementation of the ALICE detector setup.

## 4. Local Space Charge Distortions

In 2015, significant local spatial distortions were observed. These distortions were found to be caused by space charge accumulation originating from the gap between two adjacent readout chambers. The resulting deviations of the reconstructed point coordinates

(i.e., spatial positions of the TPC clusters) are on the order of several cm as shown in Figure 9.

This is much larger than the intrinsic resolution, which is on the order of 200 μm. The local space charge distortions were corrected on average; however, their fluctuations were not fully eliminated. These fluctuations can be described as

$$
\frac{\sigma_{sc}}{\mu_{sc}} = \frac{1}{\sqrt{N_{pileup}^{ion}}} \sqrt{1 + \left(\frac{\sigma_{N_{mult}}}{\mu_{N_{mult}}}\right)^2 + \frac{1}{F_{\mu_{tot}}(r)}\left(1 + \left(\frac{\sigma_{Q_{track}}}{\mu_{Q_{track}}}(r)\right)^2\right)}, \tag{1}
$$

where $N_{pileup}^{ion}$ is the number of ion pileup events, $\frac{\sigma_{N_{mult}}}{\mu_{N_{mult}}}$ is the relative RMS of the distribution of the track multiplicity, $\mu_{N_{mult}}$ is the average track multiplicity per event, $\frac{\sigma_{Q_{track}}}{\mu_{Q_{track}}}(r)$ is the relative variation of the ionization of single tracks depending on the radius $r$, and $F_{\mu_{tot}}(r)$ quantifies the amount of tracks contributing to the fluctuations for a given volume fraction [16]. The local space-charge distortion fluctuations strongly depend on the track multiplicity.

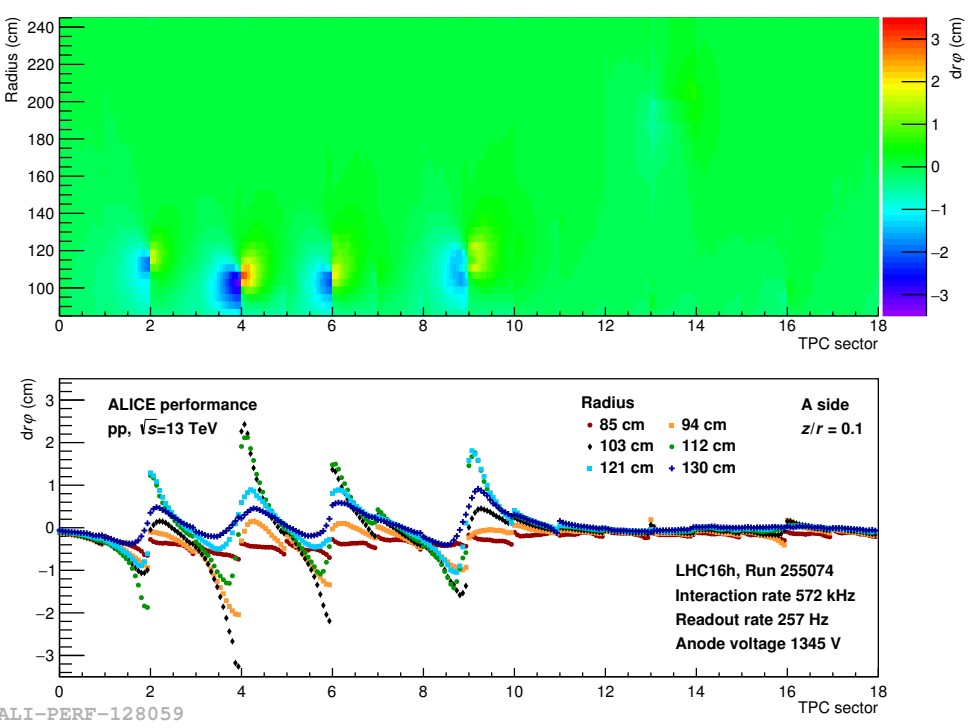

**Figure 9. Upper panel:** the deviations on the reconstructed points induced by local space charge distortions in the TPC as a function of the radius and TPC sector (which is a 20-degree wide azimuthal region) for the A side. **Lower panel:** large local deviations of up to 3 cm are observed at the TPC sectors; 2, 4, 6, 7 and 9.

To mitigate this problem, the potential at the chamber boundaries was modified to prevent electrons from entering the gaps between readout chambers before the 2018 data taking. The electron drift lines, before and after the modification, are shown in Figure 10. This change in the electric potential led to a defocusing of drifting electrons by a few millimeters at the chamber edges, thus, resulting in an efficiency loss of about 0.5–1%. Moreover, it increased the diffusion of electrons and decreased the deposited charge on the readout pads. This worsening of the charge measurements at the chamber boundaries was partially mitigated by increasing the cluster errors as explained above. Moreover, as will be discussed below, the inclusion of the TRD in tracking significantly improved the tracking performance close to the chamber boundaries.

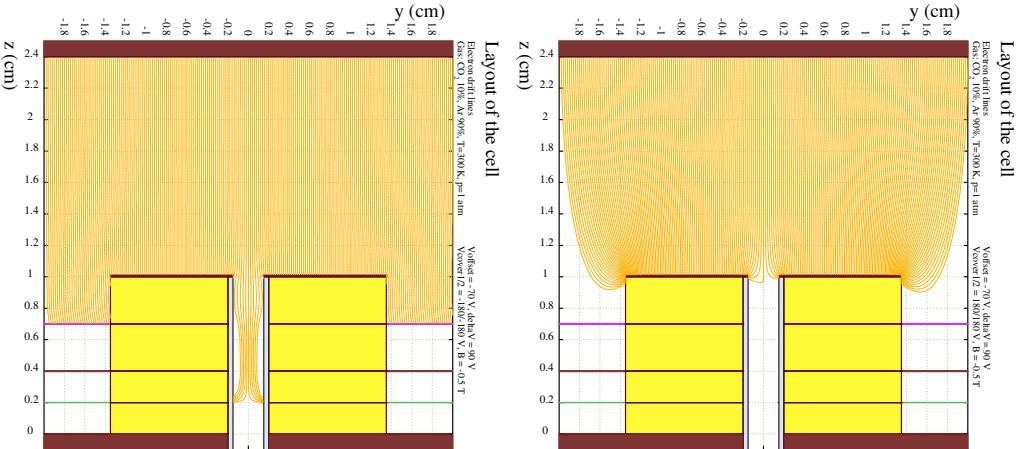

**Figure 10.** The electron drift lines, before (**left**) and after (**right**) the modification of the potential at the chamber boundaries of a TPC sector.

## 5. Using TRD in the Refit

The relative momentum resolution ($\sigma_{p_T}/p_T$) of the ALICE barrel tracking depends on several parameters. In the homogeneous detector approximation, one could write:

$$\frac{\sigma_{p_T}}{p_T} \approx p_T\,\sigma_{q/p_T} \tag{2}$$

with

$$\sigma_{q/p_T} \approx \frac{\sigma_{r\varphi}}{\sqrt{N_{cl}}}\frac{1}{BL^2} \oplus \sigma_{MS} \tag{3}$$

where $\sigma_{q/p_T}$ is the inverse transverse momentum resolution, $L$ is the track length (lever arm), $B$ is the magnetic field, $N_{cl}$ is the number of clusters, $\sigma_{r\varphi}$ is the space point resolution in the bending direction, and $\sigma_{MS}$ is a contribution due to multiple scattering.

The resolution for particles with high momentum ($p_T > 1$ GeV/$c$) is mainly determined by the track length (L) and the precision $\sigma_{r\varphi}$ of the space point measurement. Contributions due to multiple scattering are less important at high $p_T$. In combination with a significantly longer TPC+ITS lever arm ($L_{TPC} \approx 150$ cm, $L_{TPC+ITS} \approx 250$ cm) and a much better point resolution, the combined TPC– ITS tracking is significantly more accurate than TPC alone. The relative improvement in resolution at high momenta (Figure 2 left) is a factor of 3 at 10 GeV/$c$ and a factor of 6 at $p_T$ above 30 GeV/$c$.

However, the momentum resolution in combined tracking is not homogeneous; there are regions in pseudorapidity and azimuth with much worse resolution. Two typical patterns can be identified:

- The resolution is strikingly worse at the sector boundaries (Figure 11 left). At the TPC sector boundaries, the number of space points is reduced because a fraction of the particle trajectory traverses dead zones (Figure 12), and the resolution of space points $\sigma_{r\varphi}$ at the edge is also worse due to the lower gain. The resulting effect can be seen in Figure 13. For short tracks crossing the dead zone ($N_{cl} \approx 50$), the momentum resolution is 5–10-times worse than for long tracks ($N_{cl} > 150$).
- Due to distortion fluctuations in the region with high distortions (see Figure 9) the effective point resolution ($\sigma_{r\varphi}$), and thus the resolution parameters are significantly worse. The degradation of the momentum resolution due to this effect is not directly observed; instead, we show the modulation of the normalized angular resolution in Figure 14 as a function of the number of the TPC sector, which corresponds to a 20-degree wide azimuthal region.

For high-$p_T$ tracks, the inclusion of TRD in the track refit leads to an improvement in average momentum resolution by about a factor of two, as can be seen in the right panel of

Figure 2. The relative improvement depends on the interaction rate. For data collected at a higher interaction rate (affected by larger distortion fluctuations), the relative improvement due to TRD is larger. With TRD in refit, the sector modulation due to distortion fluctuations is also greatly reduced as shown in Figure 14 for pp collisions at high interaction rate (green markers compared to magenta points ).

A strong improvement is also seen in Figure 11, which shows the resolution as a function of relative position (in azimuth) within a TPC sector. When using TRD in the refit (right panel of Figure 11), the resolution improves by a factor of about 2. Note that the resolution remains the same if a given track has no hit in TRD, e.g., due to absorption in the detector material, decay before reaching the TRD or because of crossing a dead zone. The asymmetry observed close to the sector edges ($|q\Delta_{\mathrm{sector}}| > 0.4$) results from the bending direction of the tracks. Depending on the charge ($q$), the track can bend inside or outside the active area of the TRD from the dead zone (see Figure 12).

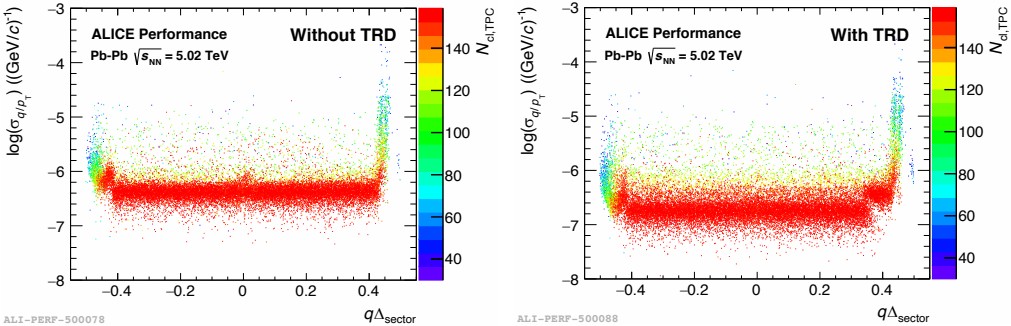

**Figure 11.** Momentum resolution for the tracks with (**right**) and without (**left**) TRD in track refit as a function of the relative position (in azimuth) inside a TPC sector (see Figure 12) for tracks with $p_{\mathrm{T}} > 5\,\mathrm{GeV}/c$.

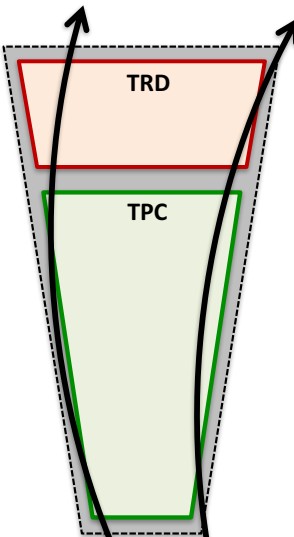

**Figure 12.** Sketch of particle tracks passing through the dead-zone (gray area) and bending inside or outside the active area.

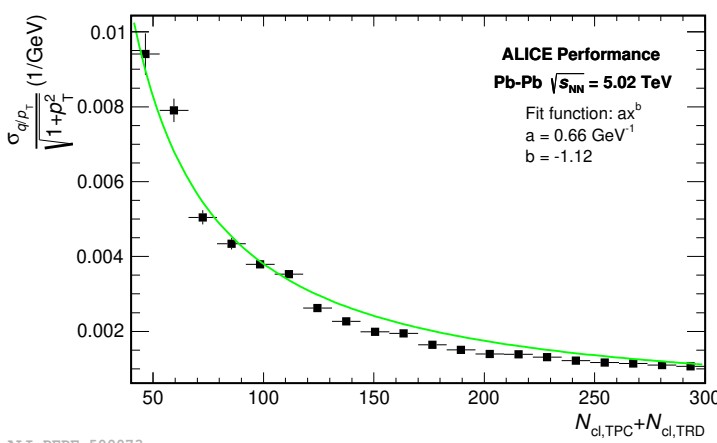

ALI-PERF-500073

**Figure 13.** Scaled combined resolution of the inverse momentum as a function of the number of TPC and TRD space points for particles with $p_T > 5\,\mathrm{GeV}/c$. The inclusion of TRD in the fit ($N_{cl} > 160$) significantly improves the momentum resolution. Relative improvement of the (TPC+TRD) resolution by a factor $\approx 2$ compared to long stand-alone TPC tracks ($N_{cl} \approx 150$) and by a factor $\approx 10$ for short stand-alone TPC tracks at the sector edges ($N_{cl} \approx 40$–$60$) (see sketch in Figure 12).

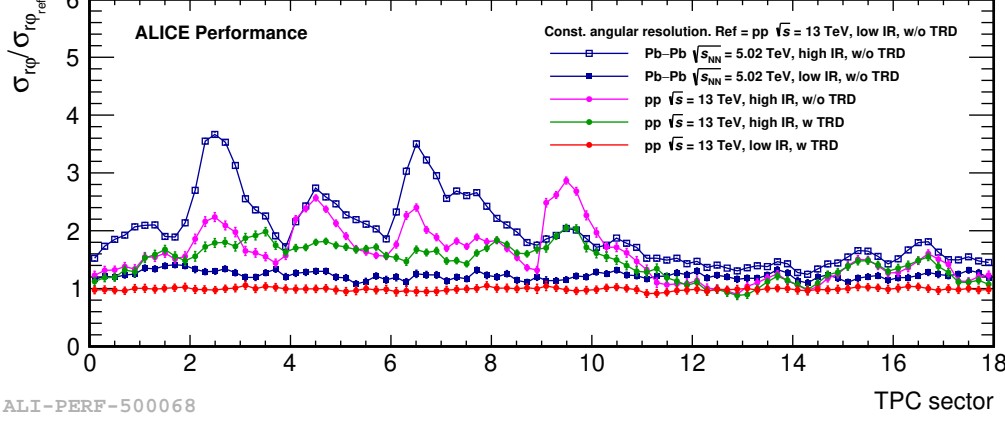

ALI-PERF-500068

**Figure 14.** The angular match between the vertex constrained TPC tracks and the ITS tracks at the primary vertex as a function of the number of TPC sector, which corresponds to a 20-degree wide azimuthal region as an approximation of the degradation of the combined tracking resolution. The data are normalized to the low interaction rate data with negligible space charge distortion fluctuations. In the sector regions 2–9 with larger distortions (see Figure 9), the angular agreement is much broader. The degradation was strongly attenuated by using the TRD in the track fit.

## 6. Conclusions

The data samples collected during the LHC Run 2 were characterized by high-track densities inside the TPC due to the high rate of pp collisions, which resulted in the pileup of several events in the TPC drift time and the high number of particles produced in Pb–Pb collisions at a center of mass energy of 5.02 TeV. A pileup of collisions in the TPC drift time was also present in about 25% of the Pb–Pb events collected in 2018, causing a further increase of the track density in the TPC volume. Calibration of the detector response in such running conditions is a challenging task for understanding and processing the data. For this, we developed the advanced software tools, "Skimmed data" and "RootInteractive framework" for a reliable and fast-turnaround optimization of the calibration and reconstruction algorithms.

The main two detector effects were the baseline fluctuations and the local space-charge distortions in the TPC, which resulted in significant deterioration in the performance of the TPC. The first is a consequence of the two characteristic features of the TPC, the

"common-mode" effect and the ion-tail, while the latter is due to space charge accumulation originating from the gap between two adjacent readout chambers.

The baseline fluctuations were initially planned to be corrected on the hardware level. However, this was not realized due to instabilities in the firmware. Therefore, an algorithm to be used offline during the raw data reconstruction was developed. Detailed signal shape studies allowed us to achieve good matching between the actual and simulated detector response. Most importantly, the bias in the $dE/dx$ due to pileup events was eliminated allowing us to recuperate more than 25% of the statistics of Pb–Pb collisions.

The local space charge distortions were mitigated by modifying the potential at the chamber boundaries to prevent electrons from entering the gaps between readout chambers. The distortions were fully eliminated with the cost of worsening of the charge measurements at the chamber boundaries. The TRD was included in the track refit to account for this problem, which significantly improved the tracking performance.

**Author Contributions:** Conceptualization, M.I., M.A. and E.H.; methodology, M.I.; software, M.I., M.A., E.H. and J.W.; formal analysis, M.I., M.A. and E.H.; investigation, R.H.M. and J.W.; data curation, R.H.M. and J.W.; writing—original draft preparation, M.I., M.A. and E.H.; writing—review and editing, M.I., M.A., E.H. and J.W.; visualization, M.I., M.A. and E.H.; All authors have read and agreed to the published version of the manuscript.

**Funding:** The ALICE Collaboration would like to thank all its engineers and technicians for their invaluable contributions to the construction of the experiment and the CERN accelerator teams for the outstanding performance of the LHC complex. The ALICE Collaboration gratefully acknowledges the resources and support provided by all Grid centres and the Worldwide LHC Computing Grid (WLCG) collaboration. The ALICE Collaboration acknowledges the following funding agencies for their support in building and running the ALICE detector: Bundesministerium für Bildung und Forschung (BMBF) and GSI Helmholtzzentrum für Schwerionenforschung GmbH, Germany; United States Department of Energy, Office of Nuclear Physics (DOE NP), United States of America.

**Conflicts of Interest:** The authors declare no conflict of interest.

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
