# Peer review of "Track Reconstruction in a High-Density Environment with ALICE"

_2571-712X, doi:10.3390/particles5010008_

Round 1
Reviewer 1 Report
The article is a very clear and well written reconstruction and performance study of the ALICE tracking detectors.
The aim of the study, that is the correction of experimental effects due to collisions pileup and high detector occupancy, is well presented both in the abstract and in the introduction. The methods developed and applied, as well as the main results, are presented in a very clear and consistent, although concise, way. The structure and organization of the article is appropriate for a publication as proceeding.
Please find below a few more specific comments and questions:
line 47: in figure 3, rather than the advantages of the skimming procedure, it's the skimming scheme itself that is depicted,
if I understand correctly. So maybe the text could be fixed ? Although of course also the advantage in terms of processing time is shown.
line 51: Are the V0's the primary vertex ? a definition added to the text would help.
line 51: it is not clear what exactly down-sampling means. If it means dropping a subset of the charged tracks based on some criterium (e.g. momentum, or number of hits ) ? Or storing less information for each track ? maybe some additional shoort detail on the criteria could be provided?
(although I understand the description is in this case intentionally brief).
line 69: a reference for the Random Forest algorithm could be added here ?
line 81: maybe you could mention first the "ion-tail" and then the "common-mode" as they are then described in this order ?
Figure 5: not sure what (Color Online) in the caption indicates ?
Maybe a sentence can be added in the caption to illustrate that the left plot is showing the ion tail of the signal ?
line 112: maybe a comment could be added, on whether the the two effects are included in the simulation, in particular at the digitization step, or not ?
line 117: the pileup interactions happening in bunch crossings after the trigger BC should not affect the signal baseline, right ? Maybe this is what is meant in the sentence but would it be possible to write that out explicitly ?
line 142: is the correction data-driven ? I.e. is the parametrization obtained comparing data with different levels of pileup to data without pileup ? A short sentence on that could be added, just for clarity.
line 149: does it mean that in order to accept more clusters, the uncertainties on the clusters are in general enlarged ?
Or only the uncertainties of the low-quality clusters are modified ?
Isn't it possible that this tuning will affect the post-fit pulls, or bias in some way the resulting track parameters ?
Figure 8: also in this case there is a (Color online) text in the caption whose meaning is not fully clear.
Eq 3: I think the MS term in the sigma_PT/PT should enter as a constant term, i.e. in eq 3 there should be also a constant term not increasing with pT. But this would be true only if the sigma_MS in eq 2 would have a 1/pT behaviour vs pT. However given that it is stated later in the text that only the pT region is considered, in which MS is negligible, one could maybe add that eq. 3 is also only valid in this approximation ? (assuming that sigma_MS in eq. 2 does not depend on pT)
line 188: maybe just a curiosity from fig. 2 it seems that the improvement is smaller than a factor 6 at 10 GeV ( 10 GeV should correspond to 0.1 in 1/pT ) ?
line 213: just out of curiosity: there is a fraction of events in Figure 13, for which the resolution remains unchanged even after the inclusion of the TRD in the fit. In particular at qDelta_sector ~ +0.4, but also over the full range
a "population" is visible in the 2D plot. Are these cases for which the association of TRD hits is failing for some reason ? Or is there some other reason for this effect ?
Maybe a comment on that could be added either in the text or in the caption of Figure 13 ?
Also, is it possible to add a simple explanation of the asimmetry of figure 3 ( both for the case without TRD and the one with TRD) ?
line 216: maybe high rate of pp data taking -> high rate of pp collisions , and then collisions -> events in the next line, would phrase the two sentences better ?
line 235: which recuperated -> allowing to recuperate
line 235: of worsening -> of a worsening
Reviewer 2 Report
The paper presents the improvements required in the event reconstruction in the ALICE's TPC in a high density environment.
General minor/stylistic comments:
Title: "in high" -> "in a high"
l.12-13: eta is normally 'pseudo-rapidity', not 'rapidity'
l.34 "Fig. 2" -> "Fig.2 (left)"
l.38 (the line after l.37, which however does not have a number): add reference to Fig2 (right).
Figure 2 right: the legend says "IR", which has never been defined in the text. Define it.
l.50 "on the one hand" -> "on one hand"
l.57 "of having a" -> "with a size"
Fig 3: The lowest box is unclear, maybe remove "Turn around time for" and replace "Time needed: "
l.69-70 and Fig.4 bottom: It is not easy/possible to see the 8 parameters clearly. Is it necessary to show them?
Fig 4: can you specify in the caption what the different colours are?
l.148 Is "compensated" the right term? Maybe "partially replaced".
Fig 9: Caption for lower plot is missing.
Equation (2) and (3): define the quantities (which resolution are you talking about in the first and second equation).
l.192: Fig 13 appears before Fig. 12
l.239 or maybe l.177: is there an efficiency loss associated to the change of the filed lines?
Fig. 12: Maybe specify in the caption the parameters for a, b in the fit that is expressed in the legend.
